# Automated single-cell proteomics providing sufficient proteome depth to study complex biology beyond cell type classifications

Claudia Ctortecka [1] ✉, Natalie M. Clark [1], Brian W. Boyle[1], Anjali Seth[2], D. R. Mani [1], Namrata D. Udeshi [1] & Steven A. Carr [1] ✉

The recent technological and computational advances in mass spectrometry-based single-cell proteomics have pushed the boundaries of sensitivity and throughput. However, reproducible quantification of thousands of proteins within a single cell remains challenging. To address some of those limitations, we present a dedicated sample preparation chip, the proteoCHIP EVO 96 that directly interfaces with the Evosep One. This, in combination with the Bruker timsTOF demonstrates double the identifications without manual sample handling and the newest generation timsTOF Ultra identifies up to 4000 with an average of 3500 protein groups per single HEK-293T without a carrier or match-between runs. Our workflow spans 4 orders of magnitude, identifies over 50 E3 ubiquitin-protein ligases, and profiles key regulatory proteins upon small molecule stimulation. This study demonstrates that the proteoCHIP EVO 96-based sample preparation with the timsTOF Ultra provides sufficient proteome depth to study complex biology beyond cell-type classifications.

Genomics, transcriptomic, and imaging methods with single-cell resolution have been shown to provide new insights to the complex cellular interplay that underlies development and disease[1–3]. Currently, most single-cell approaches aim at resolving biological heterogeneity by describing diverse sub-populations or unexpected cell types within a given sample[4,5]. In the past few years, single-cell proteomics (SCP) has developed into a viable complement to other sequencing-based omics techniques[6–11]. Mass spectrometry (MS)-based proteomics is ideally suited to study cellular identity and functionality as those are mostly driven by proteins or their post-translational modifications (PTMs)[12,13]. Until recently, the limited protein content of most mammalian cells, which is only about 50–300 pg, represented a significant challenge to MS-based SCP[14]. Even though this is over 1000 times lower than standard input of current MS-based proteomics studies, the latest advances in dedicated workflows and liquid chromatography tandem mass spectrometry (LC-MS/MS) instrumentation have begun to overcome these sensitivity limitations[15–24]. Recent SCP studies enabled the identification and quantification of thousands of proteins from a single cell in contrast to earlier reports that relied on chemical multiplexing of multiple single-cells to reach the limit of detection[7,18,25].

These strategies, however, suffer from inherent reporter ion signal suppression or ratio compression of isobaric labels, in standard data dependent acquisition (DDA), which negatively impacts quantitative accuracy[26–28]. Additionally, stochastic precursor selection in DDA results in missing data when analyzing large numbers of single cells in multiple TMT plexes[10,29–31].

Pioneering studies demonstrated reproducible in-depth profiling of one single mammalian cell at a time through optimization of sample preparation, chromatographic separation, data acquisition and data analysis[10,25,29–34]. While label-free quantification of individual cells does not suffer from interferences related to isobaric labeling, measurement throughput is significantly decreased[35]. Moreover, multiple avenues have been pursued to increase the acquisition throughput while providing the most complete proteome profiles across relatively large sample sets[23,36]. Some of these include the use of dual columns to simultaneously elute peptides of one analytical column while the other is washed and equilibrated for the subsequent sample. Additionally, high-flow and high-pressure column loading or trap-and-elute setups are used to reduce the overhead time between active peptide elution. Recently, non-isobaric labels (up to 5-plex) have been implemented in

[1]Broad Institute of MIT and Harvard, Cambridge, MA, USA. [2]Cellenion SASU, Lyon, France. ✉e-mail: cctortec@broadinstitute.org; scarr@broad.mit.edu

combination with rapid data-independent acquisition (DIA) to multiplex cells per analytical run but minimize signal interference and reduce missingness[15,37]. DIA follows a pre-defined acquisition pattern, theoretically fragmenting the same sets of precursors in every sample. To achieve the necessary acquisition speed, most DIA isolation windows are wider than in DDA methods, therefore multiplexing multiple precursors in one scan. This makes DIA, relative to DDA data, inherently more difficult to analyze. However, capitalizing on the trapped ion mobility separation (TIMS) technology of the Bruker timsTOF instruments, several SCP-DIA approaches have shown great potential[10,15,37]. Those dedicated diaPASEF strategies allow one to focus on the most productive precursor population (i.e., 400−1000 m/z), while excluding most singly charged contaminating ions.

The interpretation of convoluted DIA spectra rely on efficient chromatographic separation and reproducible sample preparation across large sample sets to minimize missing values[38–40]. Manual sample handling and transfer between reaction vessels, especially at the ultra-low input levels, can decrease reproducibility and peptide recovery, negatively impacting the ability to obtain biological meaningful data[32,41]. To eliminate manual sample handling, we benchmarked a fully automated workflow for label-free low nanoliter SCP sample preparation using the proteoCHIP EVO 96 with the picolitre and single cell dispensing robot, the cellenONE®. Similar to other successful SCP sample processing workflows like nanoPOTS[9], nPOP[42], OAD[11], the proteoCHIP 12×6[18] or plate-based methods[8], we utilize a 'one-pot approach' where all buffers and chemicals are sequentially added to the single cell. More recent developments have demonstrated that direct integration of the sample preparation vessel with the HPLC further increases reproducibility and peptide recovery[18,42–45]. However, these 'one-pot approaches' still contain all side products and excess chemicals used for sample preparation in each 'pot'. To remove the sample background efficiently and reproducibly, we have therefore seamlessly integrated the proteoCHIP EVO 96 to the high-throughput chromatography system, the Evosep One[36]. In contrast to most standard HPLC systems, the Evosep One implements disposable trap columns for in-line sample cleanup prior to LC-MS/MS analysis. This has been demonstrated to greatly increase chromatographic throughput, reproducibility, and sensitivity in conjunction with rapid MS acquisition[10,43,45,46].

Here, we combine the automated sample handling of the proteoCHIP EVO 96, high peak capacity of the nanoflow Evosep One methods and optimal ion usage of the Bruker timsTOF instruments into an integrated workflow for label-free SCP. To showcase the performance of our workflow relative to previously published methods, we demonstrate its applicability with the standard cell-line HEK-293T and recapitulate effects of LPS- treatment in hundreds of single THP-1 cells.

## Results

### The proteoCHIP EVO 96 minimizes adsorptive peptide losses

The proteoCHIP EVO 96 is a micromachined low-adsorptive PTFE chip with conical nanowells in the layout of a 96-well plate (Fig. 1a; Supplementary Fig. 1a; proteoCHIP EVO 96). For sample processing, the proteoCHIP EVO 96 is inserted into the cellenONE®. Prior to protocol initiation, 3 μL of hexadecane was dispensed in each nanowell. Lowering the chip temperature to 10 °C causes the hexadecane to solidify (melting point: 18.18 °C) within the conical nanowell (Supplementary Fig. 1b). For simultaneous lysis and digestion, we used a master-mix (MM) comprised of a MS-compatible detergent (n-Dodecyl-Beta-Maltoside (DDM), 0.2%), a protease (trypsin, 10 ng/μL) and a buffer (TEAB,

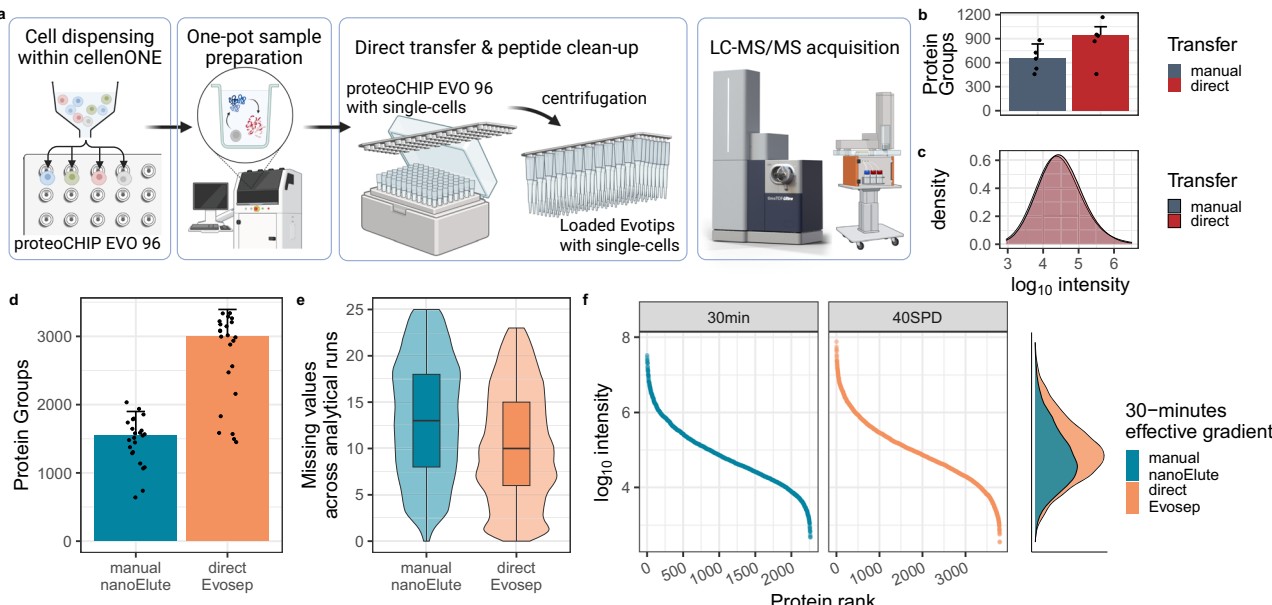

**Fig. 1 | Direct sample transfer to Evotips improves peptide recovery.**
**a** Schematic overview of the proteoCHIP EVO 96 workflow, including cell dispensing, lysis, protein digestion, transfer of single cells to Evotips and loading Evotips to the timsTOF SCP. Created with Biorender. **b** Protein group identifications by manual or direct transfer; error bar indicates Median Absolute Deviation (MAD) and **c** precursor abundance of HEK-293T single cells upon manual (gray) or direct transfer via centrifugation (red) in ddaPASEF acquisition. *N* = 10 single-cell injections prepared in one batch. **d** Protein group identifications from single-cells after manual transfer to HPLC vials with the nanoElute (blue) or direct transfer to the disposable trap columns, the Evotips (orange) acquired in diaPASEF on the timsTOF SCP. The colored bars indicate the median number of protein groups identified in the manually transferred nanoElute (blue) and directly transferred Evosep

experiments (orange). The error bar represents the MAD across all analytical runs, while the black dots indicate the number of protein groups identified per analytical run. **e** Missing quantitative values per precursor across all 25 single cells acquired on the nanoElute (blue) or the Evosep (orange) and **f** ranked log$_{10}$ intensity of single cells upon 30-minutes effective gradient on the nanoElute with manual transfer (blue) or using 40SPD on the Evosep (corresponding to 30-min effective gradient) with direct transfer. Boxplot center line indicates the median, the box limits represent the upper and lower quartiles, and the whiskers represent 1.5x the interquartile range. Source data are provided as a Source Data file. Panel a was created with BioRender.com, released under a Creative Commons Attribution-NonCommercial-NoDerivs 4.0 International license.

100 mM) as described in our previous TMT-based multiplex workflow[18]. To ensure successful deposition of the single cells within the MM droplet, we initially dispensed 150 nL MM (Supplementary Fig. 1c), followed by the isolated cell (Supplementary Fig. 1d), which was subsequently covered by another 150 nL MM (Supplementary Fig. 1e). To reduce evaporation, the SCP workflow was performed at 80% humidity within the cellenONE®. During cell lysis and protein digestion, the temperature on deck within the cellenONE® was increased to 45 °C, melting the hexadecane and fully submerging the cell in the MM (Supplementary Fig. 1f). To enforce mixing and improve cell lysis, 50 nL $H_2O$ was automatically added to the nanowells every 2 min for the full 2-h incubation time (detailed in materials and methods). All processing steps of this 'one pot' workflow were carried out within each nanowell reducing adsorptive losses due to pipetting or transfer of the sample. For direct transfer of the SCP sample to the autosampler, the proteoCHIP EVO 96 was designed to fit on top of a box of Evotips. To retain the hexadecane within the proteoCHIP EVO 96 while transferring the single cell sample to the Evotips, the sample volume is increased to a final 3 µL with 0.1% FA 1% DMSO and becomes exposed above the hexadecane layer (Supplementary Fig. 1g). Cooling the proteoCHIP EVO 96 again to 10 °C causes the oil to re-solidify, entirely separating the liquid sample droplet from the solid hexadecane. Subsequently, the cooled chip was inverted to fit each nanowell (i.e., one SCP sample) on top of each Evotip, which are disposable trap columns for in-line desalting within the Evosep One HPLC system (Fig. 1a; Supplementary Fig. 1h). Through centrifugation, the digested peptides in solution were transferred to the tips while retaining the solidified hexadecane within each of the nanowells and the now empty proteoCHIP EVO 96 was removed from the Evotip box. All Evotips were then washed manually within the centrifuge and then directly placed on the Evosep One for chromatographic separation (Fig. 1a). This workflow can be performed utilizing a full proteoCHIP preparing 96 single cells at a time or only a partial chip, depending on the experimental requirements. Based on the automated recently released full automation of Evotip loading the last step of Evotip washing can be further automated using the OT-2 liquid handler[47].

To evaluate the impact of sample transfer on peptide recovery, we initially carried out one proteoCHIP EVO 96 workflow and then transferred 5 cells by manually pipetting cell samples to Evotips, while another 5 cells were transferred via centrifugation as described above (Fig. 1a). We then analyzed those samples on the timsTOF SCP with a ~30-min effective nanoflow gradient (100 nL/min) on the Evosep One (the Whisper 40 samples per day (SPD) method by Evosep). The manually transferred samples yielded a median of 582 protein groups per analytical run, while the direct loading of the Evotips resulted in a median of 812 proteins per single cell. This indicated that automated transfer of the sample increased protein groups by 29% in comparison to the manually pipetted ones (Fig. 1b). Additionally, while no peptides with specific hydrophobicity index (GRAVY) were lost with pipetting, the direct transfer recovered lower abundant peptides (Fig. 1c, Supplementary Fig. 2a)[48]. Moreover, the unique peptide sequence overlap between the two sample transfer strategies is 99% with only 37 unique peptides in the no-transfer sample (Supplementary Fig. 2b). This demonstrates the impact of peptide adsorption at single-cell input levels and the benefit of direct sample transfer through the dedicated design of the proteoCHIP EVO 96.

Next, we further optimized the timsTOF acquisition method based on the default tryptic ddaPASEF acquisition strategy used in Fig. 1b, c. We adapted a low-input diaPASEF method to the 4 sec wide Full-Width-Half-Maximum (FWHM) elution peaks obtained with 40SPD using the IonOpticks Aurora 15 cm columns, on average resulting in 7 points across the peak. To benchmark the improvement in performance relative to the standard nanoElute LC system, we prepared single cells with the proteoCHIP EVO 96 workflow, manually transferred 25 cells to HPLC vials for 30-min effective gradient on the

nanoElute or directly loaded 25 cells to Evotips, as described above, for 40SPD (Fig. 1a). The standard nanoElute separation on the same column yielded 1,547 proteins on average per single cell (Fig. 1d). Using the modified diaPASEF acquisition method, together with the 40SPD method, we identified 3000 proteins per single HEK-293T cell (Fig. 1d). This highlighted that the combination of direct loading of samples onto Evotips together with the 40SPD Evosep One separation method recovered about 2-fold more proteins and increased sample throughput by 30% in contrast to the 30-min effective gradient on the nanoElute (i.e., 30-min effective gradient plus 20 min loading and column equilibration).

Lastly, we evaluated the differences in data completeness and dynamic range obtained using the direct Evosep- versus the manual nanoElute method for proteoCHIP EVO 96-based single-cell samples (Fig. 1e, f). For this we compared the number of missing quantitative values per precursor across all 25 single cells (i.e., analytical runs) acquired on the Evosep or the nanoElute. We observed 12% greater data completeness using the directly transferred Evosep samples compared to the manual nanoElute approach (Fig. 1e). This paralleled the loss of low abundant peptide signals after manual sample transfer and the decreased protein identifications with nanoElute separation (Fig. 1c–e). In addition to increased peptide recovery and reduced missingness, the directly transferred Evosep samples showed a similar dynamic range compared to the manual nanoElute ones (Fig. 1f). Using the manual nanoElute or the direct Evosep method, SCP samples prepared with the proteoCHIP EVO 96 spanned close to 5 orders of magnitude with similar peptide moieties (Fig. 1f, Supplementary Fig. 2c, d). Interestingly, in contrast to ddaPASEF acquisition, precursor abundance acquired with diaPASEF was higher in the direct Evosep SCP samples compared to the manual nanoElute (Fig. 1c vs. f). This is in contrast to the broader nanoElute peaks with an average FWHM of 5.1 seconds, which results in more co-eluting peptides and more convoluted spectra. Based on this, we hypothesize that the increased speed of the dedicated diaPASEF acquisition strategy in combination with the high peak capacity of the 40SPD allowed us to sample precursors more efficiently across the entire dynamic range (Fig. 1f).

## Doubling throughput still yields 3500 proteins

Most standard SCP projects require the acquisition of hundreds to thousands of single cells to resolve tissue or population heterogeneity and to understand the underlying biology. However, shorter chromatographic separation time challenged acquisition speed and effective ion usage of the timsTOF SCP. Therefore, for subsequent experiments we implemented the timsTOF Ultra, which is equipped with a 'brighter' ion source and a higher-capacity TIMS cartridge[49]. The combination of the larger inner diameter glass capillary and concomitant increased gas flow into the instrument allows for more efficient ion usage compared to the timsTOF SCP. This was reflected by the identification of over 3500 proteins per single cell with the 40SPD method, which is 15% higher compared to single cells acquired on the timsTOF SCP (compare Fig. 1d with Fig. 2a). For our comparative studies, we combine only single cells without the usage of a carrier sample for data analysis to eliminate the possibility of identification transfer without controlling the false discovery rate across the entire dataset[50,51].

Next, to realistically carry out studies including hundreds or thousands of single cells, we aimed at doubling the acquisition throughput using a 15-minute effective gradient nanoflow Evosep method. This dedicated high-throughput method on the Evosep enables analysis of 80 samples per day at 100 nL/min in conjunction with the 5 cm IonOpticks Aurora Rapid column. To minimize technical variability for comparing the separation lengths, we performed one proteoCHIP EVO 96 SCP workflow, acquired 48 single cells with the high-throughput 80SPD method, replaced the 5 cm Aurora Rapid column with the 15 cm Aurora Elite column and resumed acquisition of the remaining 48 single cells using the 40SPD method on the timsTOF

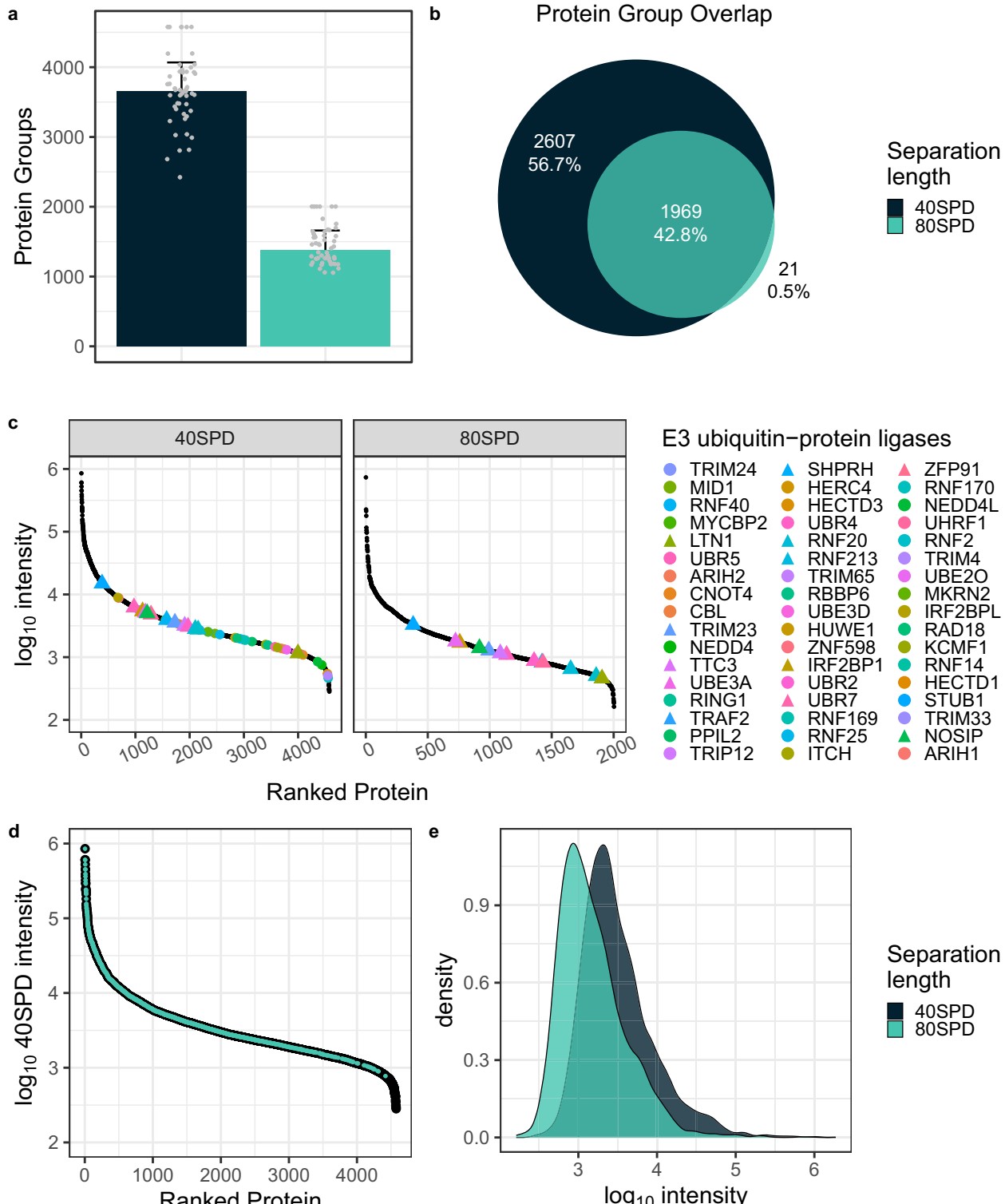

**Fig. 2 | Doubling sample throughput reduces proteome depth but still identifies biologically relevant proteins. a** Protein group identifications of single cells prepared with the proteoCHIP EVO 96 and acquired on the timsTOF Ultra at 40SPD (black; $n = 48$) or 80SPD (turquoise; $n = 48$) in diaPASEF. $N = 96$. The colored bars indicate the median and the error bar the MAD of unique protein groups identified. Gray dots indicate the number of unique protein groups identified per analytical run. **b** Protein group identification overlap of single cells acquired with 40SPD (black) or 80SPD (turquoise). **c** Ranked $\log_{10}$ intensity of proteins identified in 40SPD and 80SPD. Colored points represent 51 biologically relevant E3 ubiquitin-protein ligases sorted by their intensity in 40SPD. Targets of interest that are recovered in both 40SPD and 80SPD are indicated with a triangle. **d** $\log_{10}$ intensity of single cells acquired with 40SPD (black) overlaid with proteins also identified with 80SPD (turquoise). **e** Density distribution of $\log_{10}$ intensity of single cells acquired at 40SPD (black) or 80SPD (turquoise) in diaPASEF on the timsTOF Ultra. 'Source data are provided as a Source Data file'.

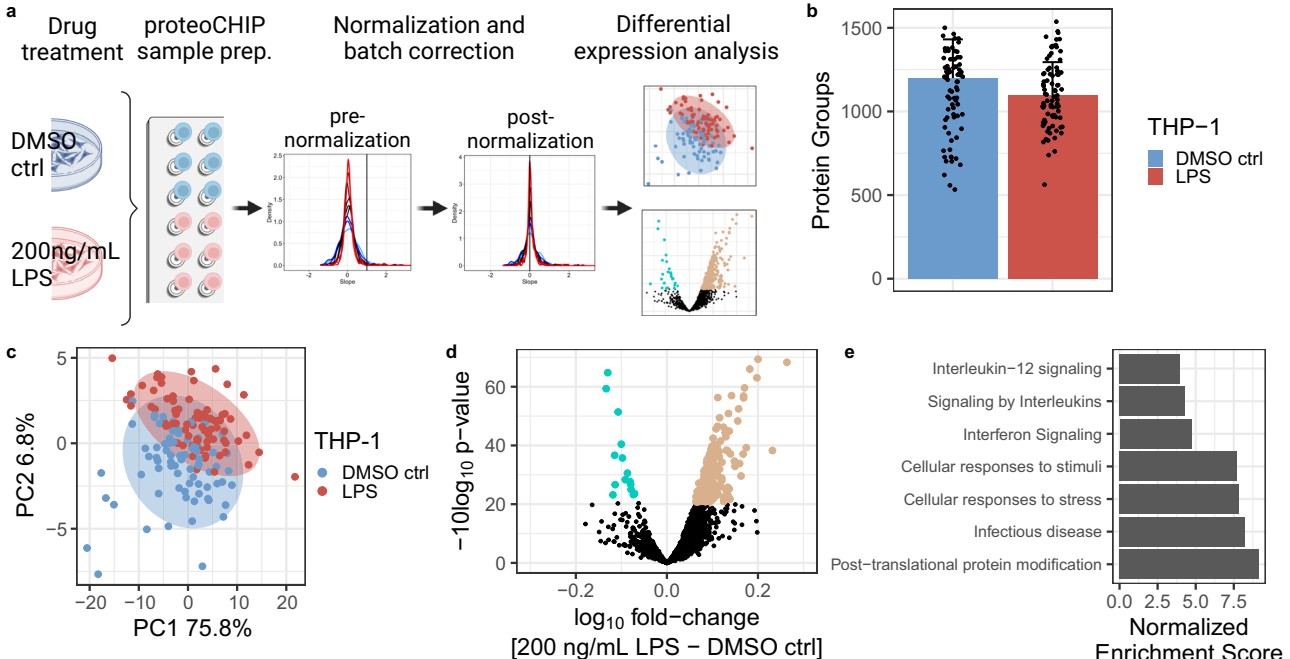

**Fig. 3 | proteoCHIP EVO 96 SCP sample preparation reflects expected proteome changes in THP-1 cells upon LPS treatment. a** Schematic overview of the proteoCHIP-based workflow, including LPS treatment of THP-1 cells, unsupervised cluster normalization and batch correction followed by differential expression analysis. **b** Protein group identifications and **c** PCA of single cells prepared with the proteoCHIP EVO 96 and acquired on the timsTOF Ultra at 40SPD (30 min effective gradient) upon 12 h of 200 ng/mL ($n = 84$) LPS or DMSO vehicle control (n = 77) treatment. The colored bars represent the median and the error bar the MAD of protein groups identified. Ellipse represents 95% confidence interval. **d** Volcano plot of moderated two-sided, two-sample $t$-test results (200 ng/mL LPS over DMSO vehicle control). Log$_{10}$ fold change and −10log$_{10}$ p-value are shown, significantly regulated proteins with an adjusted $p$-value ≤ 0.05 are indicated in blue (down, 15 proteins) and brown (up, 214 of proteins). **e** Normalized Enrichment Scores (NES) of significantly enriched Reactome pathways identified by Gene Set Enrichment Analysis (GSEA) on two-sided, two-sample $t$-test results. All shown pathways are statistically significant with adjusted $p$-value < 0.01. Source data are provided as a Source Data file. Panel a was created with BioRender.com, released under a Creative Commons Attribution-NonCommercial-NoDerivs 4.0 International license.

Ultra. As expected, with the short 80SPD separation method, the median number of proteins identified decreased to 1200 per single cell, representing a drop of ~60% compared to 40SPD (Fig. 2a). Both, 40SPD and 80SPD separate peptides at 100 nL/min, the 2-fold increased throughput with 80SPD using the 5 cm Aurora Rapid column therefore results in similar chromatographic peak widths of 4.6 versus 3.8 s, respectively. Importantly, nearly all the proteins identified with the 80SPD were also identified in the 40SPD method, which demonstrates the reproducibility of our SCP workflows and acquisition strategies (Fig. 2b).

Next, we investigated the overall abundance of quantified proteins using both the 40SPD and the 80SPD methods. The increased throughput using the 80SPD separation method reduced the dynamic range of single-cell analysis by one order of magnitude (Fig. 2c). We sought to understand the impact of this reduction on protein identification and quantification with a focus on E3 ubiquitin-protein ligases, a protein family of high biological interest. These ligases are highly diverse key players in cell cycle, signaling and cellular homeostasis balancing health and disease[52–55]. We here specifically focus on E3 ubiquitin-protein ligases due to their high dynamic range within our HEK-293T SCP datasets. Specifically, using the 40SPD method, we identify over 50 E3 ubiquitin-protein ligases in single cells while 13 are recovered using the 80SPD method (Fig. 2c). While the intensity of the ligases detected using the 80SPD method were reduced by close to one order of magnitude, the rank order of abundance remained the same between the two methods (Fig. 2c). Based on the reduced number of identifications and dynamic range using the 80SPD method, but the almost complete overlap of proteins identified, we speculated that the proteins we did not observe were of lower abundance. Indeed, proteins with an MS1 abundance of less than 1e3 in

40SPD generally dropped below the limit of detection in 80SPD (Fig. 2d). Additionally, more frequent co-isolation of precursors in shorter gradients increases the complexity of MS/MS scans and can result in signal suppression of lower abundant precursors (Fig. 2e). This demonstrates that 2-fold higher single-cell analysis throughput is feasible, at decreased analysis depth. Therefore, the experimental design must take into consideration whether the acquisition of hundreds or thousands of single cells are required for statistical significance and if the proteins of interest can still be reliably quantified to address a specific biological question.

## LPS-induced proteome changes are validated in single cells

Lipopolysaccharide (LPS) is known to stimulate inflammation, produce cytokines and activate metabolic responses in a wide range of cells[56–58]. We aimed to evaluate whether our proteoCHIP EVO 96-based SCP workflow can be used to profile the effects of LPS stimulation at the single-cell level. For this, we treated a commonly used human leukemia monocytic cell line (THP-1) with 200 ng/mL of LPS ($n = 77$) and a DMSO vehicle control ($n = 84$) for 12 h. The exposure of THP-1 cells to LPS is known to induce inflammatory cytokines, activate the tumor necrosis factor (TNF) pathway and radical oxygen species production[57,59]. The treated cells were processed with our proteoCHIP EVO 96 workflow, transferred to Evotips and data were acquired with the 40SPD method on the timsTOF Ultra (Fig. 3a). The LPS treatments were performed in process duplicates with 48 single cells per condition and proteoCHIP EVO 96, each starting from a distinct THP-1 cell passage. The two batches were processed with separate but identical proteoCHIP EVO 96 workflows and acquired on the timsTOF Ultra on different days (Supplementary Data 1). We identified up to 1537 protein groups with a median of 1149 protein groups per single cell (Fig. 3b). The nominal

decrease in identifications per single cell compared to the HEK-293T cells displayed in Fig. 2a suggests that this is due to the 2-fold decreased cell size (Fig. 2a, Fig. 3b; Supplementary Fig. 3).

Compared to bulk data, SCP datasets exhibit distinct considerations for normalization, due to technical variability throughout the sample processing, or differences in cell size[60,61] (Supplementary Fig. 3; HEK-293T – 26.7 μm, THP-1 – 13.7 μm). Thus, we normalized our SCP data using SCnorm[62], which is an approach originally developed for single-cell transcriptomics studies (Fig. 3a; Supplementary Fig. 4a, b). Normalized single-cell abundances were then $\log_{10}$-transformed. After normalization, we evaluated what proportion of the variance could be due to technical effects by processing the cells on different days using PCA regression (see Methods) and observed that approximately 9.28% of the variance can be explained by this batch effect (Supplementary Fig. 5a). We therefore batch-corrected our data using the limma package[63] to reduce this technical variability while conserving biological effects (see Methods, Supplementary Fig. 5b). PCA on the normalized, batch-corrected data shows a separation between the two treatment groups across the first (PC1, 75.8% explained variance) and second principal components (PC2, 6.8% explained variance; Fig. 3c). We identified 214 proteins as significantly upregulated in the LPS-treated cells while 15 were significantly downregulated compared to the DMSO vehicle control (Fig. 3d; moderated two-sample t-test, 229 significant proteins, adj. p-value ≤ 0.05). To evaluate the biological significance of these proteins, we performed Gene Set Enrichment Analysis (GSEA)[64] on the MSigDB Reactome gene sets (Supplementary Data 2). The top positively enriched pathways upon LPS treatment modulate proteins implicated in Interferon signaling as well as inflammatory pathways such as Interleukin signaling and specifically the Interleukin-12 family, which is consistent with the known role of LPS[65,66] (Fig. 3e).

Additionally, we compared our results to a previous in-depth proteomics analysis of bulk THP-1 cells treated with LPS[57]. Mulvey and co-authors profiled the effects of LPS treatment across a 24-hour time course and identified 4882 proteins. We found 26% of those quantified proteins also in our single cell THP-1 dataset, while the bulk study identified 3009 additional proteins (Supplementary Fig. 6a). We validated concordant fold-changes in protein expression of key response proteins that are significantly different in the LPS-treated cells compared to the DMSO vehicle control within both datasets (Supplementary Fig. 6b). As expected, we find elevated expression levels of the inflammation response to bacterial infection marker CD44 in the LPS-treated cells (Supplementary Fig. 6b)[57,67,68]. Such as, the nucleolar RNA helicase 2, DDX21, which is significantly upregulated in both the bulk and single-cell dataset. DDX21 was recently shown to be necessary for ENPP2 recruitment and therefore NF-kB mediated inflammatory response (Supplementary Fig. 6b, c)[57,69]. This and the regulation of other key LPS response proteins corroborates previous findings that even though fold changes between bulk and single cell proteomics datasets do not scale, the biological regulation correlates (Supplementary Fig. 6b)[17].

## Discussion

We present an automated label-free SCP workflow through the combination of a proteoCHIP design and sample preparation, high-throughput chromatography, dedicated acquisition methods and latest computational advances. The commercially available proteoCHIP EVO 96 SCP workflow is automated within the cellenONE® with minimal user guided operations. The temperature and humidity-controlled incubation and all nanoliter pipetting steps are performed within the cellenONE® via automated scripts. This aims at making efficient SCP workflows more accessible to the general proteomics audience and diverse core facilities. Moreover, the direct interface with the disposable trap columns, the Evotips, allows for sensitive and reproducible chromatographic separation with standardized protocols. We

demonstrate that including a single manual sample transfer step reduces protein identifications by 29%, and those losses are increased to over 49% when a standard HPLC vial is used for sample injection compared to the Evotip. This highlights the importance of transfer-free sample cleanup, especially after one-pot sample preparation to further minimize detrimental adsorptive losses. Subsequently, the Evotips are washed through the manual addition of Solvent A, which could be further automated based on the recently published automated Evotip loading protocol using the OT-2 liquid handling robot[47]. The highly reproducible peptide separation of the standardized Evosep results in high peak capacity and enables fast cycling through relatively wide diaPASEF isolation windows of 25 Th (detailed window placement in Supplementary Data 3). The 25% reduced FWHM of 40SPD on the Evosep compared to a 30-minute gradient on the nanoElute chromatographically separates peptide species more efficiently. This improves formation of distinct ion packages within the TIMS funnel based on their collisional cross section and results in less convoluted MS/MS scans with identical isolation windows (i.e., 25 Th). Moreover, improved ionization and ion transfer provided with the brighter ion source of the timsTOF Ultra yields 25% more protein identifications per single cell compared to the timsTOF SCP.

This combination of a reproducible, automated workflow with highly sensitive acquisition allowed us to characterize effects of LPS at the single-cell resolution. For this we used the monocyte cell line THP-1, which are 2-fold smaller in cell diameter compared to HEK-293T cells (i.e., 13.7 versus 26.7 μm, respectively). The reduced cell diameter parallels with lower protein identification per single THP-1 cell, as expected. We demonstrate that our proteoCHIP EVO 96 workflow is sufficiently sensitive to recapitulate previously described effects of LPS treatment in these cells. Significantly up-regulated pathways after LPS treatment included pro-inflammatory responses and post-translational modifications of proteins, which are in line with the known function of LPS. Additionally, our workflow reproduces modulations in key LPS-responsive proteins identified in previous studies. Importantly, along with sample preparation capabilities and instrument sensitivity, the cell size highly impacts proteome depth.

To aid the wider adoption of SCP, we provide detailed methods for both the timsTOF SCP and the timsTOF Ultra, clarifying the impact of peptide separation and acquisition parameters that are crucial to generate quantitative data at high confidence. We highlight that many SCP projects call for increasing the measurement throughput, however, shorter chromatographic gradients directly scale with decreased sensitivity, which retrospectively might impact the ability to make biologically relevant conclusions. We demonstrate that protein identifications are reduced by close to 60% when decreasing the gradient length by 50%. This additionally reduces the overall protein abundance and decreases detection limits by an order of magnitude. As expected, the majority of proteins identified in the 80SPD method are also identified using the 40SPD diaPASEF acquisition method, with comparable relative quantification. We demonstrate that single HEK-293T cells acquired on the timsTOF Ultra with the 40SPD chromatographic separation yield up to 4000 protein groups and on average 3500 protein groups spanning 4 orders of magnitude in abundance. This allows us to confidently quantify biologically relevant proteins, such as over 50 E3 ubiquitin-protein ligases, 13 of which are also recovered in the 80SPD method. Therefore, the overall abundance of a protein of interest in a cell line and tissue specific context should be evaluated using publicly available resources such as the Human Cell Atlas[70]. Depending on the expected abundance of the protein or proteins of interest, paired with the overall hypothesis, a combinatorial approach of two methods presented here might be most promising. While some projects require in-depth profiling of a few cells, to identify proteins of interest, it is necessary to balance acquisition time and peptide adsorption during prolonged storage. Other projects might therefore benefit from reduced depth but rapid acquisition of hundreds or even

thousands of single cells using 80SPD. The latter project might benefit from the addition of non-isobaric labels to further increase the throughput, while preserving the proteome depth of 80SPD. The promising combination of multiplexed single cells acquired at 80SPD with minimal missing quantitative values, will drive the application of SCP to more diverse biological questions. This could be especially helpful when profiling sample sets in multiple batches with large inherent biological variability, such as patient material.

Lastly, we adopt normalization approaches previously used for single-cell transcriptomics to minimize technical variability and highlight biological heterogeneity. Importantly, the normalization approach used here does not require a priori information regarding sample identities. Additionally, data analysis across all samples has been performed with a publicly available pan-human library, overcoming the need to generate an experiment-specific library for every project. We consider this specifically important in the analysis of unknown sub-populations or cell-types, which might not be accurately reflected in dedicated libraries. Similarly, the analysis of single cells in combination with higher input samples might impact identification accuracy due to the lack of false discovery rate filtering post-match between runs. In this case, identifications are heavily influenced by the representation of sub-populations within higher input samples. Moreover, we combine the normalization approach with limma-based batch correction that does not utilize or require knowledge of sample grouping. We consider this critical, as most single cell experiments lack information on sample grouping prior to data analysis. The initial separation based on technical variability highlighted the sample clustering based on treatment groups rather than experimental processing.

In conclusion, we have developed and thoroughly evaluated a fully automated label-free SCP sample preparation workflow, composed of commercially available components, that can be implemented by the community using the detailed methods provided. Deep proteomics data acquisition and analysis with high reproducibility was obtained by directly connecting efficient and reproducible chromatography with dedicated MS instrument acquisition parameters and single-cell normalization approaches. The optimization of each step within this SCP workflow allows us to achieve biologically relevant depth and realistic throughput for comprehensive proteome analysis at single cell resolution.

## Methods

### Cell culture and small molecule treatment

HEK-293T cells were cultured at 37 °C and 5% $CO_2$ in Dulbecco's Modified Eagle's High Glucose media supplemented with 10% Fetal Bovine Serum. Cells were detached with extensive phosphate-buffered saline (PBS) resuspension, cells were pelleted and washed for a total of 3 times. Cells were counted using the Countess™ 3 FL Automated Cell Counter (SN: AMQAF2000, Invitrogen) and strained using a 5-mL cell strainer tube (SN: 352235, Falcon®) to a final concentration of 200 cells/μL for optimal cell dispensing on the cellenONE®. For the proteome perturbation experiment, HEK-293T cells were seeded in a 6-well plate (Costar® 6-well plates, 3506, Corning Incorporated) at a density of $3 \times 10^5$ cells/well. THP-1 cells were seeded in a T25 flask at a density of $4 \times 10^5$ cells/mL in Gibco RPMI media supplemented with 10% Fetal Bovine Serum, Penicillin-Streptomycin and Glutamax. After 8-h, media was supplemented with LPS (final conc. of 200 ng/mL in DMSO) or DMSO vehicle control at the identical final concentration compared to LPS in DMSO (i.e., 0.05% DMSO in the LPS and the DMSO vehicle control). Treatment was carried out for 12-h prior to cell pelleting, washing, and counting. This was followed by cell straining and subsequent sample processing within the cellenONE. HEK293T cells were acquired from ATCC (Cat. Nr. CRL-3216TM), THP-1 cells were provided from the Hacohen lab at the Broad Institute. All cell lines used in this study were tested negative for mycoplasma contamination.

### Sample preparation

The proteoCHIP EVO 96 is a micromachined PTFE plate in the size of a 96-well plate with elevated nanowells to fit on top of an Evotip box, as illustrated in Fig. 1. For all single-cell experiments the conical proteoCHIP EVO 96 wells are manually prefilled with 3 μL of Hexadecane, which is solidified during subsequent sample processing at 10 °C within the cellenONE®. In detail, the proteoCHIP EVO 96 is inserted into the cellenONE® using a dedicated holder (C-PEVO-96-CHB, Scienion, Berlin, Germany), 150 nL of master mix (0.2% DDM (D4641-500MG, Sigma-Aldrich, USA/Germany), 100 mM TEAB (17902-500 ML, Fluka Analytical, Switzerland), 10 ng/μL trypsin (Promega Gold, V5280, Promega, USA)) is dispensed into each well, cells that match the isolation criteria (gated for cell diameter min. 25 μm and max. 35 μm, circularity 1.1, elongation 1.8) are dispensed to each well, which is followed by a second iteration of 150 nL of master mix dispensed to each well at 80% humidity. The temperature on deck is increased to 45 °C during lysis and enzymatic digestion, while continuously rehydrating the samples with 50 nL $H_2O$ at 2-minute intervals for 2 h. Subsequently, sample volume is increased with 3 μL 0.1% formic acid (FA) and 1% Dimethylsulfoxide (DMSO) to simultaneously quench the enzymatic reaction and increase the sample droplet size within the hexadecane. 1% DMSO is added to the sample, as low concentrations have shown to be beneficial in low-input peptide recovery[35]. Subsequently, the proteoCHIP EVO 96 is transferred to a cooling block for the Hexadecane to solidify again, resulting in complete separation of the sample droplet and the oil before centrifuging to the desired vessel (i.e., 96-well plate or Evotip box) at 335 × g for 1 min. For this, the Evotips are prepared according to manufacturer's instructions, briefly 20 μL of 0.1% FA in acetonitrile (ACN - Solvent B) is added to the Evotips using a multi-channel pipette and centrifuged at 84 × g for 1 min, the Evotips are soaked for 1-2 min in 1-propanol, 20 μL 0.1% FA (Solvent A) is added to each Evotip and centrifuged at 149 × g for 1 min. For sample loading 15 μL of Solvent A is added to each Evotip, the proteoCHIP EVO 96 is inverted and aligned on top of the Evotips. The Evotip box with the inverted proteoCHIP EVO 96 is then centrifuged at 335 × g for 1 min to transfer the sample droplet to the tips. The proteoCHIP EVO 96 is removed and visually inspected that the solidified Hexadecane remained within the nanowells and only the sample droplet was transferred to the Evotips. Subsequently, 150 μL of Solvent A is added to each Evotip and centrifuged at 149 × g for 1 minute to wash the Evotips. Solvent A is then added to the Evotip box to submerge all Evotips and prevent drying of the C18 material. Alternatively, if a 96-well plate (SureSTART™ WebSeal™ 96-Well Microtiter Plate, SN: 60180-P210B, Thermo Fisher Scientific) was used for injection, after SCP sample preparation and oil solidification on the cooling block, the proteoCHIP EVO 96 is inverted and aligned on top of the 96-well plate sample wells. The proteoCHIP EVO 96 and the 96-well plate are taped together and centrifuged at 335 × g for 1 min. Subsequently, the proteoCHIP EVO 96 is inspected for successful removal of the sample droplet from the nanowells and the presence of residual Hexadecane within the nanowells. The wells are then sealed with a silicone mat (SureSTART WebSeal 96-Well Plate Sealing Mats, 60180-M210, ThermoFisher) and stored at −20 °C until injection.

### LC-MS/MS

Samples were measured on a timsTOF SCP or timsTOF Ultra (Bruker Daltonics Gmbh) with a reversed phase nanoElute (Bruker Daltonics Gmbh), or Evosep One (Evosep) as indicated. Peptides were separated on the integrated emitter column IonOpticks Aurora Elite (15 cm × 75 μM, 1.7 μm particle size and 120 Å pore size; AUR3-15075C18-CSI) or Rapid (5 cm × 75 μM, 1.7 μm particle size and 120 Å pore size; AUR3-5075C18-CSI). For both nanoElute, peptides were eluted over a 15-minute gradient ranging from 0-20 Solvent B (0.1% FA in ACN) over 12 minutes and from 20 to 37 in 2 min, at a flow rate of 200nL/min. On the Evosep we used the standard 40SPD or 80SPD methods.

Full MS data were acquired in a range of 100–1700 m/z and 1.3 1/K0 [V·s/cm$^{-2}$] to 0.6 1/K0 [V·s/cm$^{-2}$] in diaPASEF. DIA windows range from 400 m/z to 1000 m/z with 25 Th isolation windows and were acquired with ramp times of 100 ms or otherwise indicated. The collision energy was ramped as a function of increasing mobility starting from 20 eV at 1/K0 [V·s/cm$^{-2}$] to 60 eV at 1.6 1/K0 [V·s/cm$^{-2}$]. High sensitivity detection for low sample amounts was enabled without diaPASEF data denoising.

## Data analysis

diaPASEF data was analyzed using DIA-NN v1.8.1[71] implemented on the computational platform Terra (https://app.terra.bio/) against a human fractionated library (24-fractions) acquired on a timsTOF Pro2 provided by Bruker Daltonics with standard parameters (containing a total of 301,643 precursors and 10,886 protein groups). Briefly, the protease was set to 'Trypsin/P' with one allowed missed cleavage, no allowed variable modifications and oxidized methionine. Peptide length ranges from 7 to 30 amino acids at precursor charge 1–4 within a m/z range of 300–1800 and fragment ion m/z range of 200–1800. The protein groups and precursors were filtered for 1% FDR, without MBR at robust LC parameters and retention time-dependent cross-run normalization. Post processing was performed in R, briefly, individual workflow batches were searched separately in DIA-NN and retrospectively combined using their unique Uniprot Accession number and filtered for common laboratory contaminants. Peptide level analysis, including peptide sequence overlaps are based on unique peptide sequences (Supplementary Fig. 2b, d). We consider proteins that have only shared peptides and that cannot be unambiguously identified by unique peptides as a protein group. Proteins in each group are quantified together for all subsequent experiments[71].

Data completeness for benchmarking experiments detailed in Fig. 1 was calculated on the precursor level across the sample set. GRAVY scores were calculated for every distinct, unique peptide sequence identified from the respective condition, based on the Amino Acid Hydropathy Scores[48].

Single-cell samples were grouped using unsupervised (hierarchical) clustering and then normalized using SCnorm[62]. SCnorm groups together proteins that have similar dependence of expression on proteome coverage/depth using quantile regression and then estimates scaling factors within each group of proteins. Additionally, SCnorm allows for the specification of conditions or groups of samples prior to the quantile regression step, such that this normalization approach is applied to each group separately. To group cells, we used an unsupervised clustering approach from the scran package which is based on hierarchical clustering[72]. This clustering groups together cells with similar protein expression patterns prior to the application of SCnorm. The normalized values from SCnorm were log$_{10}$-transformed and then batch corrected using limma (removeBatchEffect)[63]. Drug effects were incorporated into removeBatchEffect as a treatment effect, such that any drug effects were preserved during batch correction (Supplementary Fig. 5). Batch and drug effects were quantified by fitting a linear model between the top 10 principal components and the effect of interest and then calculating the cumulative variance explained across the top 10 principal components (Supplementary Fig. 5). Proteins were filtered for at least 30% data completeness (up to 70% missing values) across all single cells prior to statistical testing.

Moderated two-sided, two-sample t-tests were performed on the normalized, batch-corrected data using ProTIGY v1.1.7. (https://github.com/broadinstitute/protigy). An adjusted p-value of 0.05 (Benajmini-Hochberg) was used as the cutoff for statistical significance. Fold changes are reported as log$_{10}$-fold changes since the data were originally log$_{10}$-transformed (Supplementary Data 1). Gene Set Enrichment Analysis (GSEA) was performed on the signed log$_{10}$ p-values from the two-sided, two-sample t-test using ssGSEA 2.0 (https://github.com/broadinstitute/ssGSEA2.0) on the Reactome gene sets (https://www.gsea-msigdb.org/gsea/msigdb/human/genesets.jsp?collection=CP:REACTOME) (Supplementary Data 2). Reactome pathways with adjusted p-value < 0.01 were considered statistically significant (Supplementary Data 1-2).

We validated effects of LPS treatment in THP-1 cells with a published bulk dataset at the protein abundance level (Supplementary Fig. 6a–c). For this the log$_{10}$ fold changes of both datasets were scaled to allow for a relative comparison (Supplementary Fig. 6b). In Supplementary Fig. 6c we additionally overlay the distribution of the nominal p-values to highlight the comparability and similarity of observed protein regulation.

## Reporting summary

Further information on research design is available in the Nature Portfolio Reporting Summary linked to this article.

## Data availability

The mass spectrometry-based proteomics data generated in this study have been deposited in the MassIVE database under accession code MSV000093867. Source data are provided with this paper.

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

## Acknowledgements

We thank all members of our laboratory for helpful discussions. We specifically want to thank Keith Rivera, Shankha Satpathy, Hasmik Keshishian and Michael Gillette for helpful input on the manuscript. We would like to thank Jonathan Krieger, Alvaro Sebastian Vaca Jacome, Diego Assis and Matt Willetts from Bruker Daltonics GmbH and Nicholas Bache and Dorte Bekker-Jense from Evosep for their continuous support and essential input to this work. C.C. is a recipient of a SPARC Award from the Broad Institute of MIT & Harvard (#800444) that partially supported this work. This work has been supported in part by grants P01CA206978 to S.A.C from the NIH, U24CA270823, U01CA271402 to S.A.C. and U24-CA271075 to D.R.M. from National Cancer Institute (NCI) Clinical Proteomic Tumor Analysis Consortium program and from the Dr. Miriam and Sheldon G. Adelson Medical Research Foundation to N.D.U. and S.A.C.

## Author contributions

B.B. and C.C. prepared and acquired samples. A.S. conceptualized and designed the proteoCHIP. N.M.C. and C.C. performed data analysis. D.R.M., N.D.U., S.A.C. supervised the research, and revised the manuscript.

## Competing interests

The authors declare the following competing interests, A.S. is an employee of Cellenion. S.A.C. is a member of the scientific advisory boards of Kymera, PTM BioLabs, Seer and PrognomIQ. The remaining authors declare no competing interests.
