## [Peer Review File · Nature Communications]

Reviewers' Comments:

Reviewer #1:

Remarks to the Author:

Thank you for the opportunity to read and review "Automated single-cell proteomics providing sufficient proteome depth to study complex biology beyond cell type classifications". This study is a nice addition to the growing body of single cell proteomics methods. Several technologies are applied here including a new preparation and cleanup method for single cell sorting and prep, a newly released mass analyzer, and the application of downstream tools using in other fields to aid in analyzing these data.

I believe this is a fitting study for the journal in question and only have minor comments I would like addressed.

My only major comment is that it appears that considerable information is missing to set up the diaPASEF LCMS methods themselves for the 2 instruments. It should be possible to export the diaPASEF exclusion windows (both mass stop start and IMS stop start) from the instrument methods. Providing those tables as part of the source data or supplemental material would allow others to fully reproduce the LCMS settings.

Minor comments:

Page 4 first complete paragraph: Some clunky grammar, "the nanoPosts", "we here benchmark"

Page 5: Please write out the annoyingly long name of the detergent since this is the first time it appears in the text.

Figure 1: I think I see what 1e is describing, but it took a while to figure out. Could this be clarified a bit better? Also, clarifying how proteins/protein uniqueness /protein counts were determined in 1b,1c, 1d would be helpful (are these groups?)

Page 6 ("40 SPD Whisper"?)

Page 7: Last paragraph, the peak capacity here is an interesting aspect and could probably use a little extra discussion.

Figure 3b: Same clarification of proteins numbers here, please

Figure 3c: I think the mountain things on the PCA plot are interesting to look at, but I do not know what they are. Please elaborate on what this adds to the analysis

Page 11: I think some additional discussion on the LIMMA based batch correction is warranted here.

Page 14: This might be where the discussion mentioned above would fit better? It looks like a very thoughtful use of these programs was employed, it would be nice to see the logic highlighted some

Page 16: I am unclear where the Vanquish NEO system was utilized in this study.

Page 17: This may be for later in the review process, but I think you're supposed to permanently publish your Githubs to lock them from alteration.

ECR REVIEW REPORT

Thank you for the opportunity to read and review "Automated single-cell proteomics providing sufficient proteome depth to study complex biology beyond cell type classifications". This study presents a novel label-free SCP sample preparation workflow that may lay the groundwork for future significant biological discoveries using SCP. My major comments refer to some clarification in experimental approach and analysis (see below). Minor comments refer to formatting and grammar, including sentence structure and clarity.

Page 2. Abstract: A comma should be added after "limitations," i.e. "To address some of those limitations, we present..."

Page 4. Introduction: A comma should be added after "reproducibly," i.e. "To remove the sample background efficiently and reproducibly, we have therefore seamlessly..."

Page 4. Introduction: A comma should be added after "here," i.e. "Here, we combine the automated sample handling..."

Page 5. Results: This sentence appears to be incomplete. The grammar should be corrected to the following (or something similar), "Through centrifugation, the digested peptides in solution were

transferred to the tips, and the now empty proteoCHIP EVO 96..." A comma after "centrifugation" and the word "and" were added.

Page 6. Figure 1.d: Change the y-axis to be more descriptive, i.e. protein identifications or protein groups. Furthermore, it is unclear if the peptides that were identified and thus assigned to a protein were unique or proteotypic.

Page 6. Figure description: The sentence "Bar indicates median and error bar the MAD" is unclear and should be clarified for grammar and sentence structure.

Page 6. It is addressed that there were "no peptides with specific hydrophobicity index were lost with pipetting." Please address if there were changes in unique peptide identifications between methods.

Page 7. Comma missing after "this" i.e. "Based on this, we hypothesize..."

Page 8. Comma missing after "studies" i.e. "For our comparative studies, we combine only single cells..."

Page 8. Comma missing after "cells" i.e. "Next, to realistically carry out studies including hundreds or thousands of single cells, we aimed at doubling..."

Page 8. Comma missing after "method" i.e. "As expected, with the short 80SPD separation method, the median number..."

Page 9. Figure 2a. Change the y-axis to be more descriptive, i.e. protein identifications or protein groups. Furthermore, it is unclear if the peptides that were identified and thus assigned to a protein were unique/proteotypic.

Page 9. Figure 2. Inconsistent formatting as figure 2 is written as Fig. 2.:" whereas figure 1 is written as Fig. 1:

Page 6. Figure 1.d. Show individual points within the bar graph so spread of data is well visualized.

Page 9. Figure 2.a. Show individual points within the bar graph so spread of data is well visualized.

Page 10. Please elaborate as to why a focus was placed on ubiquitin-protein ligases specifically. There are multiple biologically relevant protein groups.

Page 10. Please clarify that the DMSO control was used as a vehicle control because the LPS was made in DMSO. Otherwise, please clarify why DMSO was used as a control. Change all references to "DMSO vehicle control" throughout.

Page 10. Please clarify the percent of DMSO used as a vehicle control. Literature reports percentages of DMSO (0.05%-2.5%) can change gene expression. Furthermore, high concentrations of DMSO can be toxic to cells, and thus alter gene expression and significance of findings.

Page 11. Figure 3. Referring to a previous comment, inconsistent formatting as figure 3 is written as Fig. 3.:" whereas figure 1 is written as Fig. 1:

Page 11. Figure 3. Referring to a previous comment, show individual points within the bar graph so spread of data is well visualized.

Figure 3. Page 11. Referring to the statement "The nominal decrease in identifications per single cell compared to the HEK-293T cells displayed in Fig. 2a is due to the 2-fold decreased cell size." While logical, please provide a citation to support the correlation between protein identifications using this method and cell size as this data appears to rely on correlation of data. Please incorporate "suggest" or "may indicate" into the statement as the data provided do not conclude a direct causation between cell size and protein identifications using this method without further experiments.

Page 11. Figure 3.e. Please include supplemental figure of genes or proteins that define the enriched pathways as analyzed by the Gene Set Enrichment Analysis (GSEA). For example, include the proteins that contributed to the enrichment of interferon signaling and cellular responses to stimuli pathways.

Page 13. Change effect to effects, i.e. "We demonstrate that our proteoCHIP EVO 96 workflow is sufficiently sensitive to recapitulate previously described effects of LPS treatment in these cells."

Methods. Cell Culture and small molecule treatment. Missing degrees symbol in "37C"

Methods. Sample preparation. Missing degrees symbol in "45C"

Methods. Data analysis. Please indicate if peptides were further filtered based on whether they were unique.

Reviewer #2:

Remarks to the Author:

Reviewer #3:

Remarks to the Author:

The authors provide the first characterization of the Bruker timsTOF Ultra for single-cell proteomics, as well as a workflow combining a 96 well substrate for sample preparation that can interface with Evotips for a semi-automated workflow. The manuscript will be of broad interest to the readers of nature communications, but there are numerous items that need to be addressed prior to acceptance. These are detailed below:

1. The optimized workflow with the timsTOF Ultra identifies close to 3500 proteins per cell.

Claiming 'up to 4000 proteins per cell is misleading. The average should be reported here.

Similarly, the throughput reported in the abstract should be 40 to 80 samples per day since the coverage suffers tremendously at 80 SPD.

2. When referring to nonisobaric multiplexing on page 3, the authors note these are up to 3-plex, but mDIA is up to 5-plex.

3. The following sentence needs to be fixed, as it seems to indicate that DIA operates in a 1-scan, 1-peptide scheme: "In contrast to DDA, only precursors that match specific selection parameters (i.e. charge state or intensity) are tightly isolated and individually fragmented according to a one scan, one peptide scheme."

4. The use of hexadecane makes no sense. Why is a flat working surface desirable? Hexadecane has a much lower density than water, so once it melts during the 45° steps, the sample should be under the hexadecane. Shouldn't the hexadecane then minimize evaporation? Why then are 3 uL of buffer added to the wells during the 2-h incubation? Won't the buffer be deposited over the hexadecane and not reach the working solution? And if the hexadecane is above the solution, how can the samples be spin-transferred to the evotips without the hexadecane? At a minimum, an experiment should be performed without hexadecane to show that adding potentially dangerous oil to the system is necessary and not detrimental.

5. In Figure 1C, the red distribution should be rendered transparent so that the blue distribution is not masked.

6. What are the flow rates at 40 and 80 SPD? Is it 100 nL/min at both flow rates?

7. In the right panel of figure 2C, it seems it would be better to list the ligases in order of highest to lowest abundance rather than alphabetical order, and then to use the triangles to show which are identified at both 40 and 80 SPD.

8. What is the median chromatographic peak width for both 40 and 80 SPD?

9. Why do ~90% of identified proteins appear to fall within a factor of 10 intensity range? This is much lower than what is typically observed. Is the quantification accurate or are abundances being somehow compressed?

10. Why are peak intensities reduced at 80 SPD? It is clear why coverage is reduced (due to more complex spectra), but not intensities. It appears that reported intensities are MS1-based (page 10). The same amount of sample is entering the MS at a similar flow rate and likely with similar peak widths. So what accounts for reduced intensity of a given protein that is detected at both 40 and 80 SPD?

11. One of the more troubling results presented in this study is that even with the high-dose treatment with LPS, the reported fold changes are very muted with even the most highly differentially expressed falling within a range of ~1.6. Contrast this to Figure 2 in Ref 56, where ~16x fold changes are readily observed. Is there something happening with the authors workflow or instrumentation that is causing significant ratio compression?

12. Again on p. 13, the claim of average coverage of 4000 proteins per cell is inaccurate.
13. On p. 16, is the addition of 150 uL of buffer to each evotip performed manually? If so, the claims of an automated workflow should be tempered.
14. The PDF readout from DIA-NN should be provided with the supporting information, as it will show points per peak at both 40 and 80 samples per day, intensities, etc.
15. A graphical representation of the MS acquisition methods should be provided in supporting information, showing precise window placement, placement of MS1 scans, etc.

Reviewer #4:

Remarks to the Author:

Claudia et al. detailed a single-cell proteomics platform featuring automated sample processing, swift liquid chromatography separation, highly sensitive mass spectrometry, and data-independent acquisition (DIA). The identification of approximately 4000 proteins from a single cell using this platform is indeed impressive. However, the manuscript lacks a demonstration of the platform's robustness and falls short in illustrating how the wealth of in-depth single-cell proteomics data generated can be leveraged to derive meaningful biological insights. A more thorough exploration of the platform's reliability and an elucidation of strategies for extracting biological information from the comprehensive single-cell proteomics dataset would significantly enhance the manuscript.

1. diaPASFE data was analyzed by DIA-NN against a human fractionated library. How this library is built? How many proteins and precursors are included in the library?
2. The precursors of DIA-NN's outputs were filtered for 1% FDR. What FDR is set for protein FDR?
3. Why did authors use the library-based method rather than the library-free approach? Please compare these two methods. Based on the paper (10.1021/acs.jproteome.1c00899), the comprehensive spectral library provides much more missing values of precursors than library-free method.
4. The concluding section of the paper addressing 'LPS-treated THP1 cells' appears to be relatively superficial and simplistic. I am skeptical that the identification of approximately 1000 proteins per cell can yield meaningful biological insights into LPS stimulation. Further elaboration or a more in-depth analysis may be necessary to derive substantial conclusions from the provided data.

We thank all reviewers for their exceptionally positive feedback, constructive comments, and their very helpful suggestions all of which we have used to improve the clarity of the manuscript. To address all the reviewers' comments, we have reworked multiple sections of the main text and the supplementary figures. These changes are detailed in a point-by-point reply below:

Reviewer #1 (Remarks to the Author):

Comment 1:

"Thank you for the opportunity to read and review "Automated single-cell proteomics providing sufficient proteome depth to study complex biology beyond cell type classifications". This study is a nice addition to the growing body of single cell proteomics methods. Several technologies are applied here including a new preparation and cleanup method for single cell sorting and prep, a newly released mass analyzer, and the application of downstream tools using in other fields to aid in analyzing these data. I believe this is a fitting study for the journal in question and only have minor comments I would like addressed.

My only major comment is that it appears that considerable information is missing to set up the diaPASEF LCMS methods themselves for the 2 instruments. It should be possible to export the diaPASEF exclusion windows (both mass stop start and IMS stop start) from the instrument methods. Providing those tables as part of the source data or supplemental material would allow others to fully reproduce the LCMS settings."

We thank reviewer 1 for the positive feedback and acknowledging the impact that we hope our technology will have on the single cell proteomics field. We specifically thank the reviewer for pointing out the detailed description of the entire workflow in our extensive materials and methods section from the sample preparation to the MS acquisition and data analysis. Additionally, we consider the comment regarding the diaPASEF window placements very valuable to many readers and have therefore included the DIA windows in supplemental table 3 and materials and methods section. We now note this on page 14 as follows:

The highly reproducible peptide separation of the standardized Evosep resulting in high peak capacity allows fast cycling through relatively wide diaPASEF isolation windows of 25 Th (detailed window placement in Supplemental Table 3). (page 14)

Comment 2:

"Page 4 first complete paragraph: Some clunky grammar, "the nanoPosts", "we here benchmark""

Based on the reviewer's suggestion we have modified the grammar in paragraph 2 on page 4.

To eliminate manual sample handling, we benchmarked a fully automated workflow for label-free low nanoliter SCP sample preparation using the proteoCHIP EVO 96 with the picolitre and single cell dispensing robot, the cellenONE®. Similar to other successful SCP sample processing workflows like nanoPOTS⁹, nPOP⁴², OAD¹¹, the proteoCHIP 12*6¹⁸ or plate based methods⁸, we utilize a 'one-pot approach' where all buffers and chemicals are sequentially added to the single cell. (page 4)

Comment 3:

"Page 5: Please write out the annoyingly long name of the detergent since this is the first time it appears in the text."

We apologize for missing this abbreviation within the text and have included the full name of the detergent within the text now on page 5.

For simultaneous lysis and digestion, we used a master-mix (MM) comprised of a MS-compatible detergent (n-Dodecyl-Beta-Maltoside (DDM), 0.2%), a protease (trypsin, 10 ng/ μ L) and a buffer (TEAB, 100mM) as described in our previous TMT-based multiplex workflow¹⁸. (page 5)

Comment 4:

"Figure 1: I think I see what 1e is describing, but it took a while to figure out. Could this be clarified a bit better? Also, clarifying how proteins/protein uniqueness /protein counts were determined in 1b,1c, 1d would be helpful (are these groups?)"

We agree with the reviewer that our protein grouping for all figures has not been described in appropriate detail. We have therefore revised and clarified figure legends for main Figures 1-3 and the supplementary figures, adding the term "groups" where needed. Additionally, we have included a brief explanation of protein grouping within the main text on page 7.

The manually transferred samples yielded a median of 582 protein groups per analytical run, while the direct loading of the Evotips resulted in a median of 812 proteins per single-cell. This indicated that automated transfer of the sample increased protein groups by 29% in comparison to the manually pipetted ones (Fig. 1b). (page 7)

Comment 5:

"Page 6 ("40 SPD Whisper"?)"

We have now adapted the first mention of the 40SPD method on page 6 according to the official description on the Evosep webpage (<https://www.evosep.com/plus/whisper-40-spd/>). We have also clarified the equivalency to effective gradient duration elsewhere in text as needed.

We then analyzed those samples on the timsTOF SCP with a ~30-minute effective nanoflow gradient (100 nL/min) on the Evosep One (the Whisper 40 samples per day (SPD) method by Evosep). (page 7)

Comment 6:

"Page 7: Last paragraph, the peak capacity here is an interesting aspect and could probably use a little extra discussion."

We agree with the reviewer that the peak capacity of the two HPLCs while applying the same column is very interesting to the reader and valuable information to design future experiments. We have therefore elaborated on the importance of FWHM in this context within the discussion on page 14.

The 25% reduced FWHM of 40SPD on the Evosep compared to a 30-minute gradient on the nanoElute chromatographically separates peptide species more efficiently. This improves formation of distinct ion packages within the TIMS funnel based on their collisional cross section and results in less convoluted MS/MS scans with identical isolation windows (i.e. 25 Th). (page 14)

Comment 7:

"Figure 3b: Same clarification of proteins numbers here, please"

We have extended the description of protein groupings from comment 4 accordingly to the remainder of the manuscript (page 6 onward and Figure legends 1-3).

We consider proteins that have only shared peptides and that cannot be unambiguously identified by unique peptides as a protein group and quantified together for all subsequent experiments⁷¹. (page 19)

Comment 8:

"Figure 3c: I think the mountain things on the PCA plot are interesting to look at, but I do not know what they are. Please elaborate on what this adds to the analysis"

In agreement with the reviewer we have now removed the density plots from the PCA to improve overall clarity of the figure. (page 6)

Comment 9:

"Page 11: I think some additional discussion on the LIMMA based batch correction is warranted here."

We previously elaborated on the limma-based batch correction in the methods. We have revised this section of the methods and added some details in the main text on page 13 and 19 as indicated below:

After normalization, we evaluated what proportion of the variance could be due to technical effects by processing the cells on different days using PCA regression (see Methods) and observed that approximately 9.28% of the variance can be explained by this batch effect (Supplemental Fig. 5a). We therefore batch-corrected our data using the limma package⁶³ to reduce this technical variability while conserving biological effects (see Methods, Supplemental Fig 5b). (page 13)

The normalized values from SCnorm were log₁₀-transformed and then batch corrected using limma (removeBatchEffect)⁶³. Drug effects were incorporated into removeBatchEffect as a treatment effect, such that any drug effects were preserved during batch correction (Supplemental Fig. 5). Batch and drug effects were quantified by fitting a linear model between the top 10 principal components and the effect of interest and then calculating the cumulative variance explained across the top 10 principal components (Supplemental Fig. 5). (page 19)

Comment 10:

"Page 14: This might be where the discussion mentioned above would fit better? It looks like a very thoughtful use of these programs was employed, it would be nice to see the logic highlighted some"

We thank the reviewer for highlighting our efforts for unbiased data normalization and processing, we have therefore extended the response to comment 10 in the main text (page 13) with a detailed discussion in the discussion section (page 16).

After normalization, we evaluated what proportion of the variance could be due to technical effects by processing the cells on different days using PCA regression (see Methods) and observed that approximately 9.28% of the variance can be explained by this batch effect (Supplemental Fig. 5a). We

therefore batch-corrected our data using the limma package⁶³ to reduce this technical variability while conserving biological effects (see Methods, Supplemental Fig 5b). (page 13)

Moreover, we combine the normalization approach with limma-based batch correction that does not utilize or require knowledge of sample grouping. We consider this critical, as most single cell experiments lack information on sample grouping prior to data analysis. The initial separation based on technical variability highlighted the sample clustering based on treatment groups rather than experimental processing. (page 16)

Comment 11:

"Page 16: I am unclear where the Vanquish NEO system was utilized in this study."

We thank the reviewer for catching this error. Only nanoElute and Evosep LC systems were used in the work presented in the manuscript, and we have corrected this as shown below:

Samples were measured on a timsTOF SCP or timsTOF Ultra (Bruker Daltonics GmbH) with a reversed phase nanoElute (Bruker Daltonics GmbH), or Evosep One (Evosep) as indicated. Peptides were separated on the integrated emitter column IonOpticks Aurora Elite (15 cm x 75 μ M, 1.7 μ m particle size and 120 Å pore size; AUR3-15075C18-CSI) or Rapid (5cm x 75 μ M, 1.7 μ m particle size and 120 Å pore size; AUR3-5075C18-CSI). For both nanoElute, peptides were eluted over a 15-minute gradient ranging from 0-20 Solvent B (0.1% FA in ACN) over 12 minutes and from 20-37 in 2 minutes, at a flow rate of 200nL/min. On the Evosep we used the standard 40SPD or 80SPD methods. (page 18)

Comment 12:

"Page 17: This may be for later in the review process, but I think you're supposed to permanently publish your Githubs to lock them from alteration."

We have amended the specific versions used for both the ProTIGY and ssGSEA2.0 repositories on page 19. Closer to publication, depending on editorial requirements, we will either create GitHub releases for the code versions used in this manuscript, or deposit the code in Zenodo and include relevant DOIs.

Moderated two-sample t-tests were performed on the normalized, batch-corrected data using ProTIGY v1.1.7. (<https://github.com/broadinstitute/protigy>). (page 19)

Reviewer #2 (Remarks to the Author):

ECR REVIEW REPORT

Comment 1:

"Thank you for the opportunity to read and review "Automated single-cell proteomics providing sufficient proteome depth to study complex biology beyond cell type classifications". This study presents a novel label-free SCP sample preparation workflow that may lay the groundwork for future significant biological discoveries using SCP. My major comments refer to some clarification in experimental approach and analysis (see below). Minor comments refer to formatting and grammar, including sentence structure and clarity."

“Page 2. Abstract: A comma should be added after “limitations,” i.e. “To address some of those limitations, we present...””

We thank the reviewer for their very favorable comments highlighting the importance of our novel sample preparation workflow in combination with acquisition on the timsTOF SCP and timsTOF Ultra systems. Added comma as suggested in following:

To address some of those limitations, we present a combination of fully automated single cell sample preparation utilizing a dedicated chip within the picolitre dispensing robot, the cellenONE. (page 2)

Comment 2:

“Page 4. Introduction: A comma should be added after “reproducibly,” i.e. To remove the sample background efficiently and reproducibly, we have therefore seamlessly...””

Added comma as suggested in following:

To remove the sample background efficiently and reproducibly, we have therefore seamlessly integrated the proteoCHIP EVO*96 to the high throughput chromatography system, the Evosep One³⁶. (page 4)

Comment 3:

“Page 4. Introduction: A comma should be added after “here,” i.e. “Here, we combine the automated sample handling...””

Added comma as suggested in following:

Here, we combine the automated sample handling of the proteoCHIP EVO 96, high peak capacity of the nanoflow Evosep One methods and optimal ion usage of the Bruker timsTOF instruments into a novel integrated workflow for label-free SCP. (page 4)

Comment 4:

“Page 5. Results: This sentence appears to be incomplete. The grammar should be corrected to the following (or something similar), “Through centrifugation, the digested peptides in solution were transferred to the tips, and the now empty proteoCHIP EVO 96...” A comma after “centrifugation” and the word “and” were added.”

Through centrifugation, the digested peptides in solution were transferred to the tips while retaining the solidified hexadecane within each of the nanowells and the now empty proteoCHIP EVO 96 was removed from the Evotip box. All Evotips were then washed manually within the centrifuge and then directly placed on the Evosep One for chromatographic separation (Fig. 1a). (page 5)

Comment 5:

“Page 6. Figure 1.d: Change the y-axis to be more descriptive, i.e. protein identifications or protein groups. Furthermore, it is unclear if the peptides that were identified and thus assigned to a protein were unique or proteotypic.”

In agreement with the reviewer regarding missing clarity of peptide grouping and the display of total number of protein identifications per analytical run, we now use protein groups as label of the y-axis in Figures 1b, 1d, 2a and 3b along with the figure legends and the method section.

We consider proteins that have only shared peptides and that cannot be unambiguously identified by unique peptides as a protein group. Proteins in each group are quantified together for all subsequent experiments⁷¹. (page 19)

Comment 6:

"Page 6. Figure description: The sentence "Bar indicates median and error bar the MAD" is unclear and should be clarified for grammar and sentence structure."

The colored bars indicate the median and the error bar the median absolute deviation (MAD) of protein groups identified. (Figure legend 1, 2 and 3)

Comment 7:

"Page 6. It is addressed that there were "no peptides with specific hydrophobicity index were lost with pipetting." Please address if there were changes in unique peptide identifications between methods."

We thank the reviewer for raising the critical point of unique peptide sequence overlap between the different sample preparation and chromatographic setups. We have therefore now included two additional panels to supplemental figure 1 indicating the peptide sequence overlap and described this in the main text on page 7.

Moreover, the unique peptide sequence overlap between the two sample transfer strategies is 99% with only 37 unique peptides in the no-transfer sample (Supplemental Fig. 2b). (page 7)

Comment 8:

"Page 7. Comma missing after "this" i.e. "Based on this, we hypothesize...""

Corrected. (page 8)

Comment 9:

"Page 8. Comma missing after "studies" i.e. For our comparative studies, we combine only single cells...""

Corrected. (page 8)

Comment 10:

"Page 8. Comma missing after "cells" i.e. "Next, to realistically carry out studies including hundreds or thousands of single cells, we aimed at doubling...""

Corrected. (page 9)

Comment 11:

"Page 8. Comma missing after "method" i.e. "As expected, with the short 80SPD separation method, the median number...""

Corrected. (page 9)

Comment 12:

"Page 9. Figure 2a. Change the y-axis to be more descriptive, i.e. protein identifications or protein groups. Furthermore, it is unclear if the peptides that were identified and thus assigned to a protein were unique/proteotypic."

We thank the reviewer for indicating the lack of clarity. We are now explicit that the y-axis corresponds to protein groups Figures 1b, 1d, 2a and 3b within the reviewer's comment 5.

Comment 13:

"Page 9. Figure 2. Inconsistent formatting as figure 2 is written as Fig. 2.:" whereas figure 1 is written as Fig. 1:"

Corrected. (page 10)

Comment 14:

"Page 6. Figure 1.d. Show individual points within the bar graph so spread of data is well visualized."

We agree with the reviewer that depicting the number of protein groups identified in each analytical run, in addition to the median and the MAD per analytical run, will be helpful to the reader. We have therefore included a jitter (one point per analytical run) of protein group identifications to Figures 1d, 2a and 3b including the respective figure legends.

Black dots indicate the number of protein groups identified per analytical run. (Figure 1 legend)

Comment 15:

"Page 9. Figure 2.a. Show individual points within the bar graph so spread of data is well visualized."

We have modified Figures 1d, 2a and 3b here and response to the reviewer's comment 14. The specific change to Figure 2 is noted below:

Grey dots indicate the number of protein groups identified per analytical run. (Figure 2 legend)

Comment 16:

"Page 10. Please elaborate as to why a focus was placed on ubiquitin-protein ligases specifically. There are multiple biologically relevant protein groups."

We specifically focused on E3 ubiquitin-protein ligases as we cover many of them across a large dynamic range in both the 40SPD and the 80SPD within our HEK293 cells as indicated in Figure 2c. Based on the reviewer's suggestion, we now elaborate on the choice of protein group in the main text on page 11.

We here specifically focus on E3 ubiquitin-protein ligases due to their high dynamic range within our HEK-293T SCP datasets. Specifically, using the 40SPD method, we identify over 50 E3 ubiquitin-protein ligases in single cells while 13 are recovered using the 80SPD method (Fig. 2c). (page 11)

Comment 17:

"Page 10. Please clarify that the DMSO control was used as a vehicle control because the LPS was made in DMSO. Otherwise, please clarify why DMSO was used as a control. Change all references to "DMSO vehicle control" throughout."

We agree with the importance of DMSO concentrations to control for non-specific compound effects and the possibility of altering gene expression (as indicated in reviewer's comment 18). In our experimental setup we have used the identical concentration of DMSO for the DMSO vehicle control as we did for the LPS treatment (i.e. 0.05% DMSO). Based on the reviewer's suggestion we have now

altered Figure 3b, 3c, 3d and the respective figure legends as well as extended the experimental setup on page 11 and afterwards.

For this, we treated a commonly used human leukemia monocytic cell line (THP-1) with 200 ng/mL of LPS (n = 77) and a DMSO vehicle control (n = 84) for 12 hours. (page 11)

Comment 18:

"Page 10. Please clarify the percent of DMSO used as a vehicle control. Literature reports percentages of DMSO (0.05%-2.5%) can change gene expression. Furthermore, high concentrations of DMSO can be toxic to cells, and thus alter gene expression and significance of findings."

We agree with the reviewer that the DMSO concentration in both the treated and control sample is critical to minimize technical variation and non-biologically relevant gene expression. Accordingly, we have changed the sample descriptions in Figure 3b, 3c and 3d as well as the experimental setup on page 16.

After 8-hours, media was supplemented with LPS (final conc. of 200 ng/mL in DMSO) or DMSO vehicle control at the identical final concentration compared to LPS in DMSO (i.e. 0.05% DMSO in the LPS and the DMSO vehicle control). (page 16)

Comment 19:

"Page 11. Figure 3. Referring to a previous comment, inconsistent formatting as figure 3 is written as Fig. 3.:" whereas figure 1 is written as Fig. 1.:"

Fig. 3: proteoCHIP EVO 96 SCP sample preparation reflects expected proteome changes THP-1 cells upon LPS treatment. (page 12)

Comment 20:

"Page 11. Figure 3. Referring to a previous comment, show individual points within the bar graph so spread of data is well visualized."

We have accordingly adapted Figures 1d, 2a and 3b to the reviewers comment 14.

Comment 20:

"Figure 3. Page 11. Referring to the statement "The nominal decrease in identifications per single cell compared to the HEK-293T cells displayed in Fig. 2a is due to the 2-fold decreased cell size." While logical, please provide a citation to support the correlation between protein identifications using this method and cell size as this data appears to rely on correlation of data. Please incorporate "suggest" or "may indicate" into the statement as the data provided do not conclude a direct causation between cell size and protein identifications using this method without further experiments."

Currently we mainly rely this statement on bulk protein estimates of the two cell lines in correlation with their cell diameters measured using the cellenONE. Therefore, we agree with the reviewer that this statement needs to reflect the logical conclusion it is based on, included on page 11.

The nominal decrease in identifications per single cell compared to the HEK-293T cells displayed in Fig. 2a suggests that this is due to the 2-fold decreased cell size (Fig. 2a, Fig. 3b; Supplemental Fig. 3). (page 12)

Comment 21:

“Page 11. Figure 3.e. Please include supplemental figure of genes or proteins that define the enriched pathways as analyzed by the Gene Set Enrichment Analysis (GSEA). For example, include the proteins that contributed to the enrichment of interferon signaling and cellular responses to stimuli pathways.”

We thank the reviewer for this comment and agree that the proteins supporting the Gene Set Enrichment Analysis (GSEA) are of high relevance to the audience and the conclusions drawn in the manuscript. We have therefore included an additional supplemental Table (Supplemental Table 2) detailing the results of the GSEA and the proteins contributing to the pathways indicated in Figure 3e.

To evaluate the biological significance of these proteins, we performed Gene Set Enrichment Analysis (GSEA⁶²) on the MSigDB Reactome gene sets (Supplemental Table 2). (page 13)

Fold changes are reported as log₁₀-fold changes since the data were originally log₁₀-transformed (Supplemental Table 1). Gene Set Enrichment Analysis (GSEA) was performed on the signed log₁₀ p-values from the two-sample *t*-test using ssGSEA 2.0 (<https://github.com/broadinstitute/ssGSEA2.0>) on the Reactome gene sets (<https://www.gsea-msigdb.org/gsea/msigdb/human/genesets.jsp?collection=CP:REACTOME>) (Supplemental Table 2). Reactome pathways with adjusted p-value < 0.01 were considered statistically significant (Supplemental Table 1-2). (page 19)

Comment 22:

“Page 13. Change effect to effects, i.e. “We demonstrate that our proteoCHIP EVO 96 workflow is sufficiently sensitive to recapitulate previously described effects of LPS treatment in these cells.””

Corrected. (page 14)

Comment 23:

“Methods. Cell Culture and small molecule treatment. Missing degrees symbol in “37C””

Corrected. (page 15)

Comment 24:

“Methods. Sample preparation. Missing degrees symbol in “45C””

Corrected. (page 16)

Comment 25:

“Methods. Data analysis. Please indicate if peptides were further filtered based on whether they were unique.”

We have now included that critical information to the methods section within the main text on page 18, page 19 and the Supplemental Figure 2 legend.

Peptide level analysis, including peptide sequence overlaps are based on unique peptide sequences (Supplemental Fig. 1b, 1d). (page 18)

Data completeness for benchmarking experiments detailed in Figure 1 was calculated on the precursor level across the sample set. GRAVY scores were calculated for every distinct, unique peptide sequence identified from the respective condition, based on the Amino Acid Hydropathy Scores⁴⁷. (page 19)

Reviewer #3 (Remarks to the Author):

Comment 1:

The authors provide the first characterization of the Bruker timsTOF Ultra for single-cell proteomics, as well as a workflow combining a 96 well substrate for sample preparation that can interface with Evotips for a semi-automated workflow. The manuscript will be of broad interest to the readers of nature communications, but there are numerous items that need to be addressed prior to acceptance. These are detailed below:

1. The optimized workflow with the timsTOF Ultra identifies close to 3500 proteins per cell. Claiming 'up to 4000 proteins per cell is misleading. The average should be reported here. Similarly, the throughput reported in the abstract should be 40 to 80 samples per day since the coverage suffers tremendously at 80 SPD.

We thank the reviewer for this detailed evaluation of our manuscript with specific focus on the applicability of the protocol by the Nature Communications audience. We aim at highest transparency of the workflow and the results we report per single cell samples and have therefore amended experimental details and average protein group identifications to the abstract and main text as detailed below.

The proteoCHIP EVO 96 can be directly interfaced with the Evosep One chromatographic system for in-line desalting and highly reproducible separation with a throughput of 40-80 samples per day. (page 2)

The implementation of the newest generation timsTOF Ultra with our proteoCHIP EVO 96-based sample preparation workflow reproducibly identifies up to 4,000 protein groups with an average of 3,500 protein groups per single HEK-293T without a carrier or match-between runs. (page 2)

We demonstrate that single HEK-293T cells acquired on the timsTOF Ultra with the 40SPD chromatographic separation yield up to 4,000 and on average 3,500 protein groups spanning 4 orders of magnitude in abundance. (page 15)

Comment 2:

"2. When referring to nonisobaric multiplexing on page 3, the authors note these are up to 3-plex, but mDIA is up to 5-plex."

We highly appreciate the highlight of this mistake within the introduction and have corrected it on page 3.

Recently, non-isobaric labels (up to 5-plex) have been implemented in combination with rapid data independent acquisition (DIA) to multiplex cells per analytical run but minimize signal interference and reduce missingness^{15,37}. (page 3)

Comment 3:

"3. The following sentence needs to be fixed, as it seems to indicate that DIA operates in a 1-scan, 1-peptide scheme: "In contrast to DDA, only precursors that match specific selection parameters (i.e. charge state or intensity) are tightly isolated and individually fragmented according to a one scan, one peptide scheme."

We agree with the reviewer that this sentence was misleading in the initial introduction of the manuscript, we have rephrased on page 3 in the revised version.

DIA follows a pre-defined acquisition pattern, theoretically fragmenting the same sets of precursors in every sample. (page 3)

Comment 4:

"4. The use of hexadecane makes no sense. Why is a flat working surface desirable? Hexadecane has a much lower density than water, so once it melts during the 45° steps, the sample should be under the hexadecane. Shouldn't the hexadecane then minimize evaporation? Why then are 3 µL of buffer added to the wells during the 2-h incubation? Won't the buffer be deposited over the hexadecane and not reach the working solution? And if the hexadecane is above the solution, how can the samples be spin-transferred to the evotips without the hexadecane? At a minimum, an experiment should be performed without hexadecane to show that adding potentially dangerous oil to the system is necessary and not detrimental."

We thank the reviewer for this comment, based on which we have extended the description of the workflow including the intended use of hexadecane which is to minimize evaporation during lysis and digestion. During method development, we have repeatedly experienced complete evaporation of the 300 nL sample droplet during incubation at 45°C. This evaporation is observed despite humidity at 80% and continuous rehydration within the cellenONE during elongated incubation times. We now elaborate on the detailed sample preparation method in the main text on page 5 and added Supplemental Fig. 1.

For sample processing, the proteoCHIP EVO 96 is inserted into the cellenONE®. Prior to protocol initiation, 3 µL of hexadecane was dispensed in each nanowell. Lowering the chip temperature to 10 °C causes the hexadecane to solidify (melting point: 18.18 °C) within the conical nanowell (Supplemental Fig. 1b). For simultaneous lysis and digestion, we used a master-mix (MM) comprised of a MS-compatible detergent (n-Dodecyl-Beta-Maltoside (DDM), 0.2%), a protease (trypsin, 10 ng/µL) and a buffer (TEAB, 100mM) as described in our previous TMT-based multiplex workflow¹⁸. To ensure successful deposition of the single cells within the MM droplet, we initially dispensed 150 nL MM (Supplemental Fig. 1c), followed by the isolated cell (Supplemental Fig. 1d), which was subsequently covered by another 150 nL MM (Supplemental Fig. 1e). To reduce evaporation, the SCP workflow was performed at 80% humidity within the cellenONE®. During cell lysis and protein digestion, the temperature on deck within the cellenONE® was increased to 45 °C, melting the hexadecane and fully submerging the cell in the MM (Supplemental Fig. 1f). To enforce mixing and improve cell lysis, 50 nL H₂O was automatically added to the nanowells every 2 minutes for the full 2-hour incubation time (detailed in materials and methods). All processing steps of this 'one pot' workflow was carried out within each nanowell reducing adsorptive losses due to pipetting or transfer of the sample. For direct transfer of the SCP sample to the autosampler, the proteoCHIP EVO 96 was designed to fit on top of a box of Evotips. To retain the hexadecane within the proteoCHIP EVO 96 while transferring the single cell sample to the Evotips, the sample volume is increased to a final 3 µL with 0.1% FA 1% DMSO and

becomes exposed above the hexadecane layer (Supplemental Fig. 1g). Cooling the proteoCHIP EVO 96 again to 10 °C causes the oil to re-solidify, entirely separating the liquid sample droplet from the solid hexadecane. Subsequently, the cooled chip was inverted to fit each nanowell (i.e. one SCP sample) on top of each Evotip, which are disposable trap columns for in-line desalting within the Evosep One HPLC system (Fig. 1a; Supplemental Fig. 1h). Through centrifugation, the digested peptides in solution were transferred to the tips while retaining the solidified hexadecane within each of the nanowells and the now empty proteoCHIP EVO 96 was removed from the Evotip box. All Evotips were then washed manually within the centrifuge and then directly placed on the Evosep One for chromatographic separation (Fig. 1a). (page 5)

Comment 5:

"5. In Figure 1C, the red distribution should be rendered transparent so that the blue distribution is not masked."

We have adapted rendering in Figure 1c accordingly.

Comment 6:

"6. What are the flow rates at 40 and 80 SPD? Is it 100 nL/min at both flow rates?"

To highlighting the importance of flow rates for the run chromatographic length comparisons we have now included the flow rates of 100 nL/min for both 40SPD and 80SPD in the main text on page 7 and 9.

We then analyzed those samples on the timsTOF SCP with a ~30-minute effective nanoflow gradient (i.e. 100 nL/min) on the Evosep One (the Whisper 40 samples per day (SPD) method by Evosep). (page 7)

This dedicated high throughput method on the Evosep enables analysis of 80 samples per day (SPD) at 100 nL/min in conjunction with the 5 cm IonOpticks Aurora Rapid column. (page 9)

Comment 7:

"7. In the right panel of figure 2C, it seems it would be better to list the ligases in order of highest to lowest abundance rather than alphabetical order, and then to use the triangles to show which are identified at both 40 and 80 SPD."

We have now adapted Figure 2c and the respective figure legend accordingly.

Colored points represent 51 biologically relevant E3 ubiquitin-protein ligases sorted by their intensity in 40SPD. (page 10)

Comment 8:

"8. What is the median chromatographic peak width for both 40 and 80 SPD?"

To further elaborate on the importance of FWHM in DIA quantification and identification confidence we. The identical flow rates for both 40 and 80SPD resulted in similar peak widths of 4.6 seconds for 40SPD and 3.8 seconds for 80 SPD resulting in 4-5 points across the peak with our dedicated diaPASEF method.

Both, 40SPD and 80SPD separate peptides at 100 nL/min, the 2-fold increased throughput with 80SPD using the 5 cm Aurora Rapid column therefore results in similar chromatographic peak widths of 4.6 versus 3.8 seconds, respectively. (page 9)

Comment 9:

“9. Why do ~90% of identified proteins appear to fall within a factor of 10 intensity range? This is much lower than what is typically observed. Is the quantification accurate or are abundances being somehow compressed?”

We thank the reviewer for the in-depth evaluation of our data. We compared our intensity distribution to other published single-cell proteomics datasets (i.e. Brunner et al., MSB, 2022 or Truong et al., Angew. Chem., 2023). The intensities reported in these studies also spread across 4 orders of magnitude while the majority fall within a factor of 10, as the reviewer notes.

Comment 10:

“10. Why are peak intensities reduced at 80 SPD? It is clear why coverage is reduced (due to more complex spectra), but not intensities. It appears that reported intensities are MS1-based (page 10). The same amount of sample is entering the MS at a similar flow rate and likely with similar peak widths. So what accounts for reduced intensity of a given protein that is detected at both 40 and 80 SPD?”

We agree with the reviewer regarding the expected similar intensity between 40SPD and 80SPD at identical flow rates of 100 nL/min and similar peak widths (as indicated in comment 8), we suspect the shift in intensity due to the two different columns used for the consecutive acquisition of 80SPD and 40SPD.

Comment 11:

“11. One of the more troubling results presented in this study is that even with the high-dose treatment with LPS, the reported fold changes are very muted with even the most highly differentially expressed falling within a range of ~1.6. Contrast this to Figure 2 in Ref 56, where ~16x fold changes are readily observed. Is there something happening with the authors workflow or instrumentation that is causing significant ratio compression?”

We understand the concern of the reviewer regarding fold change comparisons between the two manuscripts. However, the bulk data acquired in ref. 57 (old ref 56) differs in overall proteome depth achieved through extensive off-line fractionation, chromatographic setup and gradient as well as instrumentation, all of which can reduce compression. Moreover, it has been previously observed by others that single-cell fold changes are compressed relative to the identical sample acquired in bulk (Specht et al., Genome Biology 2021). While we certainly agree with the reviewer that our fold-changes are compressed relative to the previous LPS bulk study, we here aim to compare relative fold changes achieved within our method as indicated in revised Supplemental Figure 6. We therefore focus on significantly changing proteins across the entire dataset for the GSEA analysis to pre-filter for proteins that are most likely of importance to the biological system analyzed. Additionally, we find that after scaling the fold changes between the two studies are highly comparable, which we attribute to the different input levels. Our single cell samples are acquired close to the limit of detection and might be more affected by ratio compression, however the comparison of nominal p-values between the two studies revealed that our study allows to highlight significant differences in those cells.

Comment 12:

"12. Again on p. 13, the claim of average coverage of 4000 proteins per cell is inaccurate."

Based on this we have adapted the phrasing throughout to state that we observe up to 4,000 protein groups per single cell with an average of 3,500 protein groups per single HEK-293T cell at 40SPD throughput. (page 15)

We demonstrate that single HEK-293T cells acquired on the timsTOF Ultra with the 40SPD chromatographic separation yield up to 4,000 and on average 3,500 protein and span 4 orders of magnitude. (page 15)

Comment 13:

"13. On p. 16, is the addition of 150 uL of buffer to each evotip performed manually? If so, the claims of an automated workflow should be tempered."

We have adapted the text as follows:

All Evotips were then washed manually within the centrifuge and then directly placed on the Evosep One for chromatographic separation (Fig. 1a). This workflow can be performed utilizing a full proteoCHIP preparing 96 single-cells at a time or only a partial chip, depending on the experimental requirements. Based on the automated recently released full automation of Evotip loading the last step of Evotip washing can be further automated using the OT-2 liquid handler⁴⁷. (page 5-6)

Subsequently, the Evotips are washed through manual addition of Solvent A, which could be further automated based on the recently published automated Evotip loading protocol using the OT-2 liquid handling robot⁴⁷. (page 14)

Comment 14:

"14. The PDF readout from DIA-NN should be provided with the supporting information, as it will show points per peak at both 40 and 80 samples per day, intensities, etc."

We agree with the reviewer that the data summary is highly useful to obtain an in-depth overview of the data presented. The Google Cloud (<https://app.terra.bio/>) based implementation of DIA-NN does not create PDF summary reports, however, we have included all summary reports to our MASSIVE upload to provide all readers with the necessary information (i.e. points per peak, intensities).

Comment 15:

"15. A graphical representation of the MS acquisition methods should be provided in supporting information, showing precise window placement, placement of MS1 scans, etc."

To enable reproducibility of our method we have included the detailed window placement to Supplemental Table 3. This table can be directly imported to timsControl and used to recreate the method.

The highly reproducible peptide separation of the standardized Evosep resulting in high peak capacity allows fast cycling through relatively wide diaPASEF isolation windows of 25 Th (detailed window placement in Supplemental Table 3). (page 14)

Reviewer #4 (Remarks to the Author):

“Claudia et al. detailed a single-cell proteomics platform featuring automated sample processing, swift liquid chromatography separation, highly sensitive mass spectrometry, and data-independent acquisition (DIA). The identification of approximately 4000 proteins from a single cell using this platform is indeed impressive. However, the manuscript lacks a demonstration of the platform's robustness and falls short in illustrating how the wealth of in-depth single-cell proteomics data generated can be leveraged to derive meaningful biological insights. A more thorough exploration of the platform's reliability and an elucidation of strategies for extracting biological information from the comprehensive single-cell proteomics dataset would significantly enhance the manuscript.”

We thank reviewer 4 for highlighting our unique combination of dedicated single-cell proteomics sample preparation, high-throughput liquid chromatography and sensitive mass spectrometry acquisition. In this initial demonstration of our technological advancement, we aim to demonstrate the workflow and allow other groups to implement the platform to their biological questions. We have therefore chosen a well described application with known biology that we recapitulated at single-cell resolution to illustrate the broad biological applicability of the proteoCHIP EVO 96 workflow. Based on the reviewer's suggestion we have nevertheless expanded the comparison to previous studies and revised Supplemental Fig. 5 (now Supplemental Fig. 6) and the result section on page 13 as detailed below.

Comment 1:

“1. diaPASFE data was analyzed by DIA-NN against a human fractionated library. How this library is built? How many proteins and precursors are included in the library?”

In agreement with the reviewer we have amended details to the human library to the method section on page 18.

diaPASEF data was analyzed using DIA-NN v1.8.1⁷¹ implemented on the computational platform Terra (<https://app.terra.bio/>) against a human fractionated library (24-fractions) acquired on a timsTOF Pro2 provided by Bruker Daltonics with standard parameters (containing a total of 301,643 precursors and 10,886 protein groups). (page 18)

Comment 2:

“2. The precursors of DIA-NN's outputs were filtered for 1% FDR. What FDR is set for protein FDR?”

We apologize for overseeing the missing information from our materials and method section and have now included the protein group FDR filter estimation on page 18.

The protein groups and precursors were filtered for 1% FDR, without MBR at robust LC parameters and retention time-dependent cross-run normalization. (page 18)

Comment 3:

“3. Why did authors use the library-based method rather than the library-free approach? Please compare these two methods. Based on the paper (10.1021/acs.jproteome.1c00899), the comprehensive spectral library provides much more missing values of precursors than library-free method.”

We agree with the reviewer that the direct comparison of library-based (including different library depths) and library-free analysis would be highly informative for the community but exceeds the scope of the current study.

Comment 4:

“4. The concluding section of the paper addressing 'LPS-treated THP1 cells' appears to be relatively superficial and simplistic. I am skeptical that the identification of approximately 1000 proteins per cell can yield meaningful biological insights into LPS stimulation. Further elaboration or a more in-depth analysis may be necessary to derive substantial conclusions from the provided data.”

We agree with the reviewer that the more than 2-fold smaller THP-1 cells challenge the achieved proteome depth at single cell resolution, however, we recover ~26% of all proteins quantified in the bulk study in our single cell dataset (Supplemental Fig. 6a). We readily demonstrate separation of the two treatment groups (LPS treated and DMSO vehicle control) along principal component 1 and 2 following treatment independent normalization and batch correction (Fig. 3c, Supplemental Fig. 5a-b). Moreover, the directionality of proteins that are significantly regulated proteins upon LPS stimulation of THP1 cells are well correlated between the two studies (Supplemental Fig. 6b). We recover highly relevant proteins involved in the expected pathways as demonstrated with the GSEA in Fig. 3e and Supplemental Fig. 6b. In this initial demonstration and benchmarking of the proteoCHIP EVO 96 we intentionally chose a well described biological application to validate our findings based on previous literature. We have now expanded on the comparison to this earlier bulk study and relevant biological findings on pages 12-13 and in the discussion on page 16.

Reviewers' Comments:

Reviewer #1:

Remarks to the Author:

Thank you for your thorough response to both my and my co-reviewer's comments to this nice new addition to the emerging field of single cell proteomics. Upon review of the reviewer response, I believe this manuscript is ready for publication.

Reviewer #2:

Remarks to the Author:

I co-reviewed this manuscript with one of the reviewers who provided the listed reports. This is part of the Nature Communications initiative to facilitate training in peer review and to provide appropriate recognition for Early Career Researchers who co-review manuscripts

Reviewer #3:

Remarks to the Author:

The authors have substantially improved the manuscript, which is close to ready for acceptance. I would prefer that the authors acknowledge the odd results mentioned in Reviewer 3 comments 9-11 in the manuscript itself and not just in the rebuttal, but this is not necessarily required.

Reviewer #4:

Remarks to the Author:

I think that the authors have adequately addressed the comments made by the reviewers in the revised version of the manuscript. Therefore, I have no further comments.